# Multiple communication mechanisms between sensor kinases are crucial for virulence in *Pseudomonas aeruginosa*

Vanessa I. Francis[1], Elaine M. Waters[2], Sutharsan E. Finton-James[1], Andrea Gori[1], Aras Kadioglu[2], Alan R. Brown[1] & Steven L. Porter [1]

Bacteria and many non-metazoan Eukaryotes respond to stresses and threats using two-component systems (TCSs) comprising sensor kinases (SKs) and response regulators (RRs). Multikinase networks, where multiple SKs work together, detect and integrate different signals to control important lifestyle decisions such as sporulation and virulence. Here, we study interactions between two SKs from *Pseudomonas aeruginosa*, GacS and RetS, which control the switch between acute and chronic virulence. We demonstrate three mechanisms by which RetS attenuates GacS signalling: RetS takes phosphoryl groups from GacS-P; RetS has transmitter phosphatase activity against the receiver domain of GacS-P; and RetS inhibits GacS autophosphorylation. These mechanisms play important roles in vivo and during infection, and exemplify an unprecedented degree of signal processing by SKs that may be exploited in other multikinase networks.

[1] Biosciences, Geoffrey Pope Building, College of Life and Environmental Sciences, University of Exeter, Exeter EX4 4QD, UK. [2] Department of Clinical Infection, Microbiology and Immunology, Institute of Infection and Global Health, University of Liverpool, Liverpool L69 7BE, UK. Correspondence and requests for materials should be addressed to S.L.P. (email: s.porter@exeter.ac.uk)

Canonical two-component systems (TCSs), comprising a single sensor kinase (SK) working with its cognate response regulator (RR)[1,2], detect and respond to stimuli but are not well suited to making complex decisions requiring the integration of multiple signals. However, multikinase networks, where several SKs collaborate to detect and integrate signals, can make these sophisticated decisions requiring the evaluation of multiple stimuli. Multikinase networks regulate processes as diverse as asymmetric cell division[3–5], sporulation[6], chemotaxis[7], nitrogen metabolism[8], stress responses[9], virulence[10,11], biofilm formation[12], and differentiation into fruiting bodies[13,14]. Many multikinase networks feature interactions between their constituent SKs, but how they affect signalling output is unclear[15–17].

Most SKs are homodimeric proteins, containing sensory domains for detecting stimuli and controlling the activity of their catalytic core, comprising the HisKA and HATPase domains. The HATPase domain binds ATP and phosphorylates a histidine residue within the HisKA domain. For simple SKs, following autophosphorylation, phosphotransfer occurs to an aspartate in the receiver (REC) domain of the RR. In more complex SKs (~20% of bacterial and ~90% of eukaryotic examples), additional phosphorylation sites, contained within either attached REC (hybrid SKs) or attached REC and Hpt domains (unorthodox SKs)[18], participate in multi-step phosphorelays, which comprise His-to-Asp and Asp-to-His phosphotransfer reactions, conveying phosphoryl groups to the RR. Phosphorylation activates the RR, mediating a response to the stimulus[19]. Signals are terminated by hydrolysis of the aspartyl-phosphate residue, which is an autocatalytic process, often augmented by either transmitter phosphatase activity of the SK or extrinsic phosphatases[20–22]. The phosphotransfer and phosphatase reactions are highly specific, ensuring fidelity of signalling[23–29].

*Pseudomonas aeruginosa* is a leading cause of healthcare-acquired infections[30,31]. It infects vulnerable patients, e.g. neonates or those with cystic fibrosis, burn wounds, or cancer. It causes acute and chronic infections[32]; acute infections (e.g. pneumonia and sepsis) feature motility and type III secretion (T3S), while, chronic infections (e.g. cystic fibrosis lung) involve biofilm production and type VI secretion (T6S)[32–34]. The GacS multikinase network plays a key role in orchestrating the transition between acute and chronic infection[35–38]. Central to this network is the unorthodox SK, GacS, which phosphorylates the RR, GacA. GacA-P activates transcription of the regulatory RNAs, RsmY and RsmZ, which sequester the translational regulator, RsmA, thereby upregulating genes required for chronic infection and downregulating acute infection[39–43].

Although only GacS-P phosphorylates GacA, other SKs influence GacA-P levels via GacS. LadS senses calcium[44], and promotes chronic infection by phosphorylating GacS[45,46]. In contrast, the hybrid SK, RetS, binds and inhibits GacS via unknown mechanisms, favouring acute infection[10,47–49]. The ligand controlling RetS is unknown, but RetS responds to lysis of kin *P. aeruginosa* cells[50]. Another SK, PA1611, sequesters RetS, relieving GacS from its inhibition[51,52].

The inhibitory interaction of RetS with GacS is considered the paradigm for negative regulation in multikinase networks[10]. Previously, it was proposed that they form an inactive heterodimer incapable of autophosphorylation[10]. Here, we demonstrate that the interactions are much more extensive, with RetS having three distinct mechanisms for downregulating GacS. All are important for RetS function and play major roles in virulence. These mechanisms represent an unprecedented level of cross-communication between SKs that can be widely utilised by other multikinase networks.

## Results

**RetS takes phosphoryl groups from GacS-P in mechanism 1.** RetS has a degenerate HATPase domain lacking the conserved G-boxes and consequently is unable to autophosphorylate. The cytoplasmic portions of GacS and RetS (denoted GacSc and RetSc) interact, and the inclusion of RetSc in GacSc autophosphorylation reactions reduces the steady-state levels of GacSc-P produced[10]; here, we determine the molecular mechanisms responsible. We began by testing whether RetSc could accelerate the dephosphorylation of purified GacSc-P (Fig. 1a, b); these reactions did not contain any residual ATP. We detected phosphorylation of RetSc (Fig. 1b), indicating phosphotransfer had occurred from GacSc-P to RetSc. To determine which of the three phosphorylation sites of RetS accepts the phosphoryl group from GacS-P, we prepared mutant RetSc proteins lacking these sites (wild-type RetSc, with its three native phosphorylation sites is denoted RetSc(HDD)). The mutant protein lacking the phosphorylation site in REC1 (D713) was unchanged in its ability to be phosphorylated by GacSc-P (Supplementary Fig. 1). We found that the mutant proteins lacking the phosphorylation site in REC2 (D858), RetSc(HDA) and RetSc(HAA), were not phosphorylated by GacSc-P (Fig. 1c, d), indicating that D858 is phosphorylated by GacSc-P.

To identify which of the three phosphorylation sites of GacSc-P was the phosphodonor for this phosphotransfer reaction, we engineered a mutant GacSc protein, GacSc(HAQ), retaining only its autophosphorylation site and lacking its REC and Hpt domain phosphorylation sites. We found that when purified GacSc(HAQ)-P was coincubated with RetSc, RetSc-P was produced and the intensity of the GacSc(HAQ)-P band was decreased, meaning that phosphotransfer had occurred from GacSc(HAQ)-P to RetSc (Fig. 1f). RetSc-P levels did not rise to the same extent as GacSc(HAQ)-P levels decreased during this phosphotransfer reaction because RetSc-P dephosphorylates and therefore RetSc-P levels are determined not only by the rate of phosphotransfer from GacSc(HAQ)-P but also by how quickly RetSc-P dephosphorylates. No phosphotransfer was seen from GacSc(HAQ)-P to the mutant RetSc proteins lacking D858, RetSc(HDA) and RetSc(HAA) (Fig. 1g, h). These data indicate that the autophosphorylatable His residue in the HisKA domain of GacS is a phosphodonor for D858 in REC2 of RetS. We refer to this as mechanism 1 (Fig. 1i).

**RetS is a transmitter phosphatase for GacS-P in mechanism 2.** Wild-type GacSc-P autodephosphorylates with a half-time of $25\pm2$ min (Fig. 1j, k). RetS mutant proteins, RetSc(HDA) (Fig. 1g) and RetSc(HAA) (Fig. 1h), lacking D858, are disabled for mechanism 1 and cannot dephosphorylate GacSc(HAQ)-P. However, they still speed up the dephosphorylation of wild-type GacSc-P (Fig. 1k); each reducing the half-time from $25\pm2$ min to $15\pm1$ min (Fig. 1k). Likewise, RetS$_{HK}$, lacking both REC domains and comprising only the catalytic core (HisKA and the degenerate HATPase domain) of RetS, also reduced the GacSc-P dephosphorylation half-time (Fig. 1j, k). This suggests RetS has a second mechanism for dephosphorylating GacS-P that resides within the catalytic core of RetS, and targets either the REC or Hpt phosphorylation sites of GacS (present in wild-type GacSc-P but not in GacSc(HAQ)-P). We examined the dephosphorylation of GacSc(HDQ)-P, which lacks the phosphorylation site in the Hpt domain, and found that RetSc(HDA) and RetS$_{HK}$ could catalyse its dephosphorylation (Supplementary Fig. 2), indicating that the Hpt phosphorylation site is not the target and therefore the target is the REC domain phosphorylation site.

The majority of SKs have kinase and phosphatase activity. While RetS has a degenerate HATPase domain rendering it

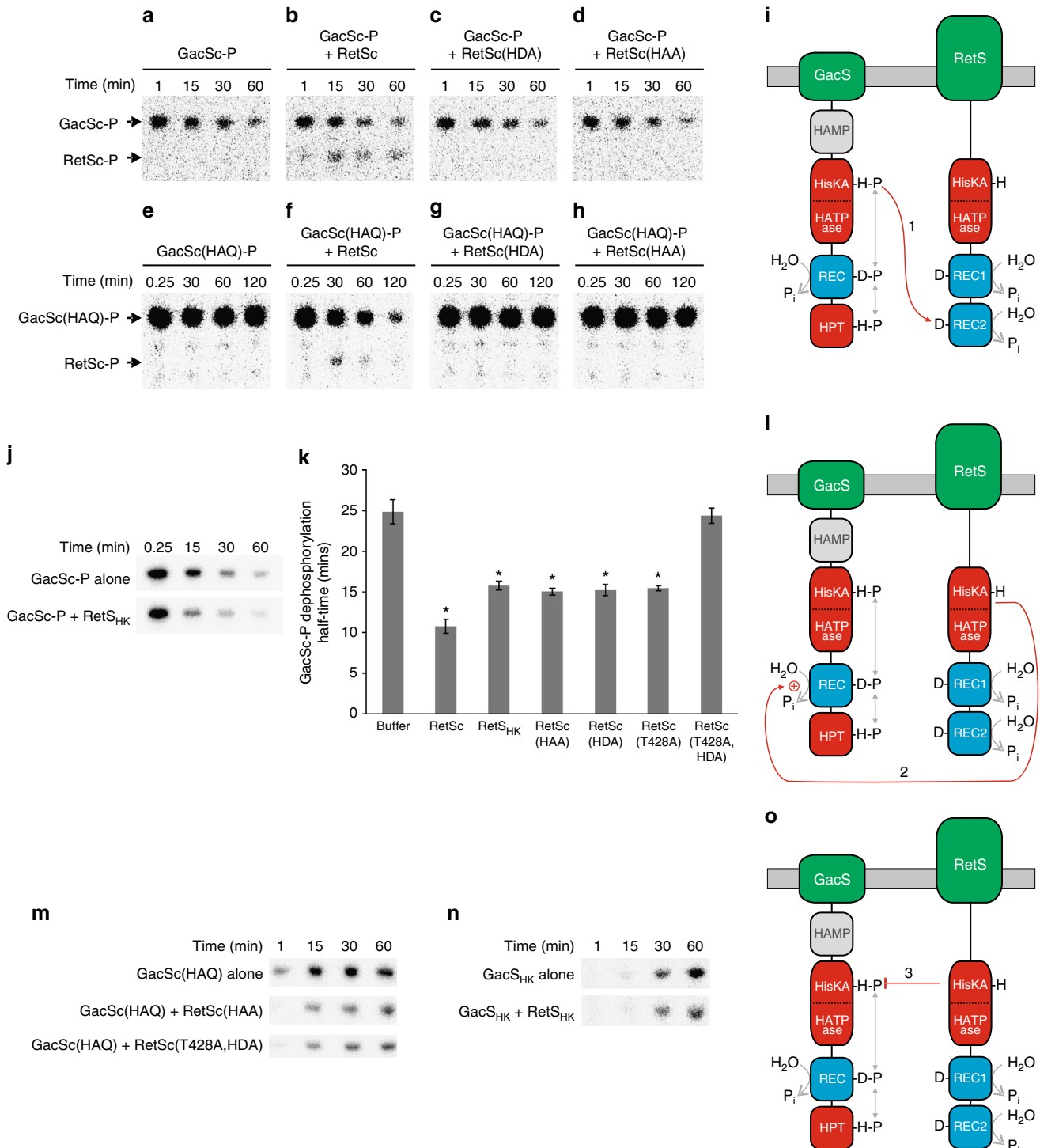

**Fig. 1** RetS downregulates GacS signalling via three distinct mechanisms. **a–i** Mechanism 1—RetS takes phosphoryl groups from GacSc-P. Phosphorimages of SDS-PAGE gels showing phosphotransfer from **a–d** GacSc-P to **a** buffer control, **b** RetSc, **c** RetSc(HDA) and **d** RetSc(HAA), and from **e–h** GacSc(HAQ)-P to **e** buffer control, **f** RetSc, **g** RetSc(HDA) and **h** RetSc(HAA). Experiments were repeated five times and a representative image shown. **i** Cartoon depicting mechanism 1, where phosphotransfer occurs from the His residue in the HisKA domain of GacS to D858 in REC2 of RetS. **j–l** Mechanism 2—RetS has transmitter phosphatase activity against GacSc-P. **j** Phosphorimages of SDS-PAGE gels measuring the dephosphorylation of GacSc-P alone (top) and with RetS$_{HK}$ (bottom). **k** GacS-P dephosphorylation half-times in the presence of various RetS mutant proteins. Error bars show SEM from eight replicates. *Significantly faster than GacSc-P autodephosphorylation ($P < 0.05$, one-way ANOVA). **l** Cartoon depicting mechanism 2, where RetS uses transmitter phosphatase activity against the REC domain of GacS-P. **m–o** Mechanism 3—RetS inhibits the GacS autophosphorylation reaction. Phosphorimages of SDS-PAGE gels comparing the autophosphorylation of: **m** GacSc(HAQ) on its own (top), with RetSc(HAA) (middle), and with RetSc(T428A,HDA) (bottom). **n** GacS$_{HK}$ on its own (top) and with RetS$_{HK}$ (bottom). Experiments were repeated eight times and a representative image shown. **o** Cartoon depicting mechanism 3, where the catalytic core of RetS blocks the autophosphorylation of GacS

incapable of autophosphorylation, its HisKA domain retains the conserved H-box including the motif (HExxT) needed for transmitter phosphatase activity[20,53]. We therefore, hypothesised that RetS has transmitter phosphatase activity directed towards the REC domain of GacS. Consistent with this, the mutant protein, RetSc(T428A,HDA), lacking both the conserved T428 residue from the HExxT phosphatase motif, and D858 required for mechanism 1, had no ability to speed up GacSc-P dephosphorylation (Fig. 1k). Therefore, mechanism 2 is enhanced dephosphorylation of GacS-P by transmitter phosphatase activity of RetS against the REC domain of GacS (Fig. 1l).

**RetS inhibits the autophosphorylation of GacS in mechanism 3.** Mechanisms 1 and 2, described above, dephosphorylate GacS-P. Here, we tested whether RetS could affect the autophosphorylation rate of GacS; we did this using mutant versions of GacSc and RetSc where both dephosphorylation mechanisms were disabled. RetSc(HAA) cannot dephosphorylate GacSc (HAQ)-P (Fig. 1h); however, the presence of RetSc(HAA) in a GacSc(HAQ) autophosphorylation reaction reduced the level of GacSc(HAQ)-P that accumulated (Fig. 1m). Likewise, RetSc (T428A,HDA) which is disabled for mechanisms 1 and 2 (Fig. 1k), inhibited the autophosphorylation of GacSc(HAQ) (Fig. 1m). Similar results were seen when using only the kinase catalytic cores (HisKA and HATPase domain) of GacS and RetS; the presence of RetS$_{HK}$ in a GacS$_{HK}$ autophosphorylation reaction reduced the level of GacS$_{HK}$-P that accumulated (Fig. 1n). This means there is a third mechanism of interaction between GacS and RetS where the kinase core of RetS inhibits autophosphorylation of GacS (Fig. 1o).

**Mechanisms 1 and 2 control biofilm formation.** To determine the relative contribution of the three mechanisms in vivo, we replaced the wild-type retS gene in the chromosome with mutant versions; the retS(HDA) mutant lacking mechanism 1, the retS (T428A) mutant lacking mechanism 2, the retS(T428A,HDA) mutant lacking mechanisms 1 and 2, and the ΔretS mutant lacking all three mechanisms. We confirmed that the mutant proteins were expressed at comparable levels to wild-type RetS by western blotting (Supplementary Note 1 and Supplementary Fig. 3).

Deletion of retS enhances biofilm formation (Fig. 2)[47,48,54]. The retS point mutants (retS(HDA), retS(T428A) and retS(T428A, HDA)) produced significantly more biofilm than the PAO1 strain (Fig. 2a). The mutant lacking mechanism 1, retS(HDA), had similar biofilm levels to the ΔretS mutant, indicating that mechanism 1 plays a major role in regulating biofilm formation. While biofilm levels were significantly higher in the retS(T428A) mutant than in the wild-type strain, they were not elevated to the same extent as the ΔretS mutant, indicating that mechanism 2 plays a significant role in controlling biofilm formation although to a lesser extent than mechanism 1. To verify that the increased biofilm formation observed in the retS point mutants was a consequence of GacS dysregulation, we introduced these mutations into a ΔgacS background. Like their parent ΔgacS mutant, these double gacS/retS mutants produced significantly less biofilm than the PAO1 strain (Supplementary Fig. 4), consistent with the retS point mutations affecting signalling via GacS. These data indicate that the control of GacS signalling by mechanisms 1 and 2 of RetS are important for regulating biofilm formation.

**Contribution of the three mechanisms to controlling RsmY&Z.** The GacS network controls the expression of two small RNAs, RsmY and RsmZ[55]. The ΔretS mutant has elevated levels of RsmY and RsmZ because of its increased GacS activity[10,55–56]. We

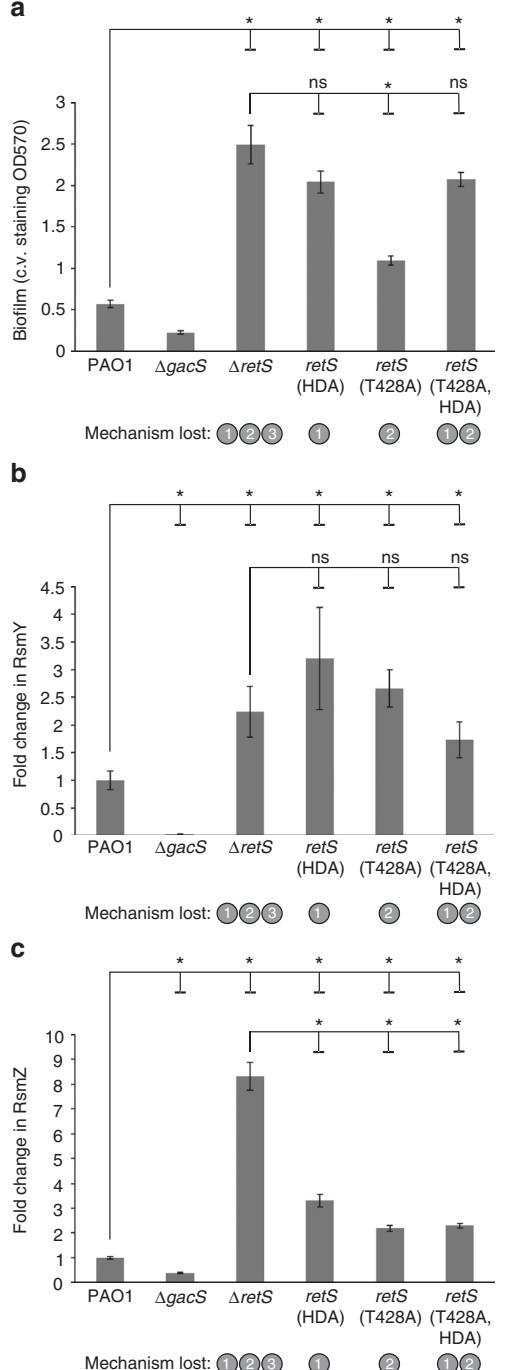

**Fig. 2** Mechanisms 1 and 2 control biofilm formation and the expression levels of RsmY, while all three mechanisms contribute to controlling RsmZ levels. **a** Quantification of biofilm formation on peg-lidded 96-well plate by crystal violet staining. Plates inoculated with the mutant strains were incubated for 10 h with shaking at 37 °C. Error bars show SEM (three biological repeats each containing five technical repeats). *Indicates significantly different comparisons ($P < 0.05$, one-way ANOVA). See also Supplementary Fig. 4 for these mutations in a ΔgacS background. **b, c** Relative expression level of RsmY (**b**) and RsmZ (**c**) in the mutant strains relative to the wild-type PAO1 strain. RNA levels were measured using qRT-PCR. Error bars show SEM (three biological repeats, with three technical repeats per biological repeat). *Indicates significantly different comparisons ($P < 0.05$, one-way ANOVA). The mechanisms disabled in each mutant strain are indicated

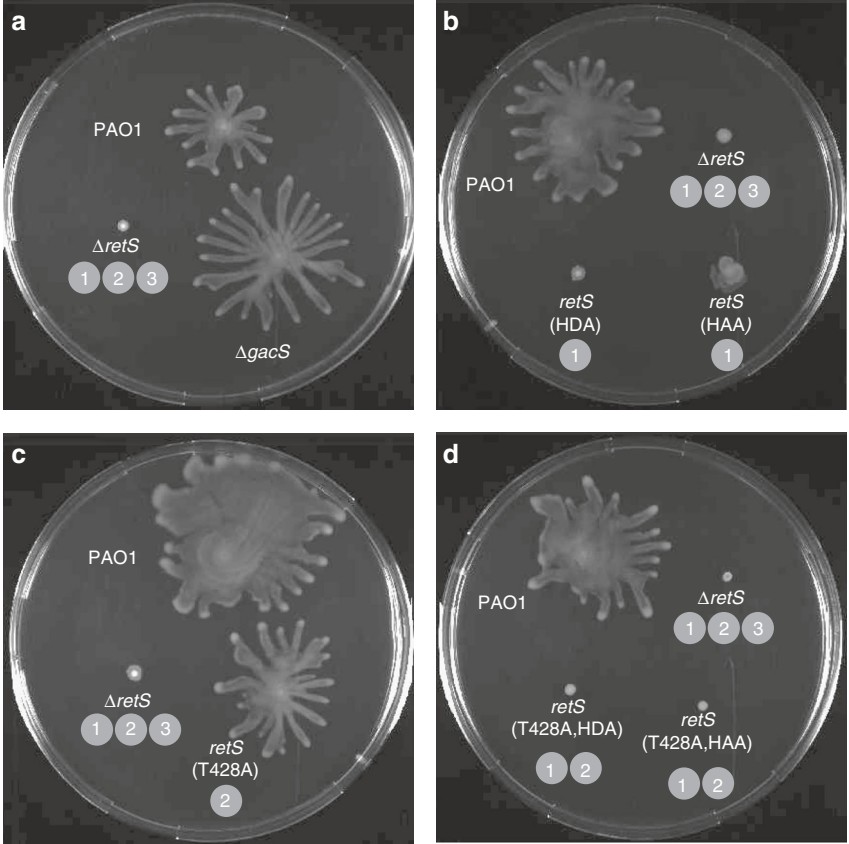

**Fig. 3** Mechanism 1 is required for swarming motility. **a–d** Representative images of the swarming of the wild-type PAO1 strain and its mutant derivatives. Experiments were repeated three times. The mechanisms disabled in each mutant strain are indicated in circles

found that levels of RsmY and RsmZ were increased in all *retS* point mutant strains lacking mechanisms 1 and 2 (Fig. 2b, c), indicating that these mechanisms play major roles in controlling *rsmY* and *rsmZ* expression. The mutant lacking both mechanisms 1 and 2 (*retS*(T428A,HDA)), however, did not produce as much RsmZ as the Δ*retS* mutant (Fig. 2c), suggesting that mechanism 3, alongside mechanisms 1 and 2, contributes a significant role to controlling the expression of *rsmZ*.

**Mechanism 1 is important for swarming motility.** Unlike the wild-type strain, PAO1, the Δ*retS* mutant did not swarm (Fig. 3a). We found that all mutants lacking mechanism 1, e.g. *retS*(HDA) (Fig. 3b), failed to swarm, indicating that mechanism 1 is essential. The mechanism 2 mutant, *retS*(T428A), did swarm, indicating that mechanism 2 is not required (Fig. 3c). Consistent with mechanism 1 being essential, the *retS*(T428A,HDA) mutant also failed to swarm (Fig. 3d).

**Virulence in *Galleria mellonella* requires mechanisms 1 and 2.** To assess the role of the three mechanisms in virulence, the *retS* mutants were tested in a *Galleria mellonella* infection model. Larvae were injected with 20–40 CFUs of PAO1 or mutant strains. The PAO1 infected larvae all died within 21 h of infection, whereas over 60% of the larvae infected with the Δ*retS* mutant survived to the end of the experiment (47 h post-infection) (Fig. 4a). Similar to the Δ*retS* mutant, the *retS*(HDA) and *retS* (T428A,HDA) mutants showed severely attenuated virulence with significantly more larvae surviving than those infected with PAO1 (Fig. 4a). The *retS*(T428A) mutant showed significantly delayed killing compared to those infected with PAO1 (Fig. 4a). We examined the phenotypes of the *retS*(HDA)Δ*gacS* and the

*retS*(T428A)Δ*gacS* mutants and found that they were as virulent as the Δ*gacS* mutant (Fig. 4b), confirming that the reduction in virulence seen in the *retS*(HDA) and *retS*(T428A) mutants was dependent on the presence of GacS and therefore a consequence of GacS dysregulation. In summary, mechanism 1 is essential for virulence in *G. mellonella* whereas loss of mechanism 2 delays killing.

**RetS overexpression allows mechanism 3 to support virulence.** The mutant lacking mechanisms 1 and 2, *retS*(T428A,HDA), phenocopies the Δ*retS* mutant in most of the above assays, with the only significant difference being that RsmZ levels are not elevated to the same extent as the Δ*retS* mutant (Fig. 2c). This indicates that mechanism 3 alone is not sufficient for RetS function, and that dephosphorylation of GacS-P, via mechanisms 1 and 2, is essential. We hypothesised that while mechanism 3 cannot compensate for the loss of mechanisms 1 and 2 at native expression levels of RetS, it may be able to compensate at higher expression levels. To test this, we overexpressed RetS(T428A, HDA), which possesses only mechanism 3, in the Δ*retS* mutant. Successful complementation was seen (Fig. 4c) indicating that, at non-physiologically high expression levels, mechanism 3 compensates for the loss of mechanisms 1 and 2. However, at physiological expression levels of RetS, this compensation cannot occur, and mechanisms 1 and 2 are both required.

**Mechanisms 1 and 2 are required for virulence in mice.** We used a mouse model of acute respiratory infection to probe the role of the mechanisms in virulence. Mice were inoculated intranasally with $2 \times 10^7$ CFUs of *P. aeruginosa*. Only 10% of mice infected with PAO1 survived beyond 28 h of infection. All

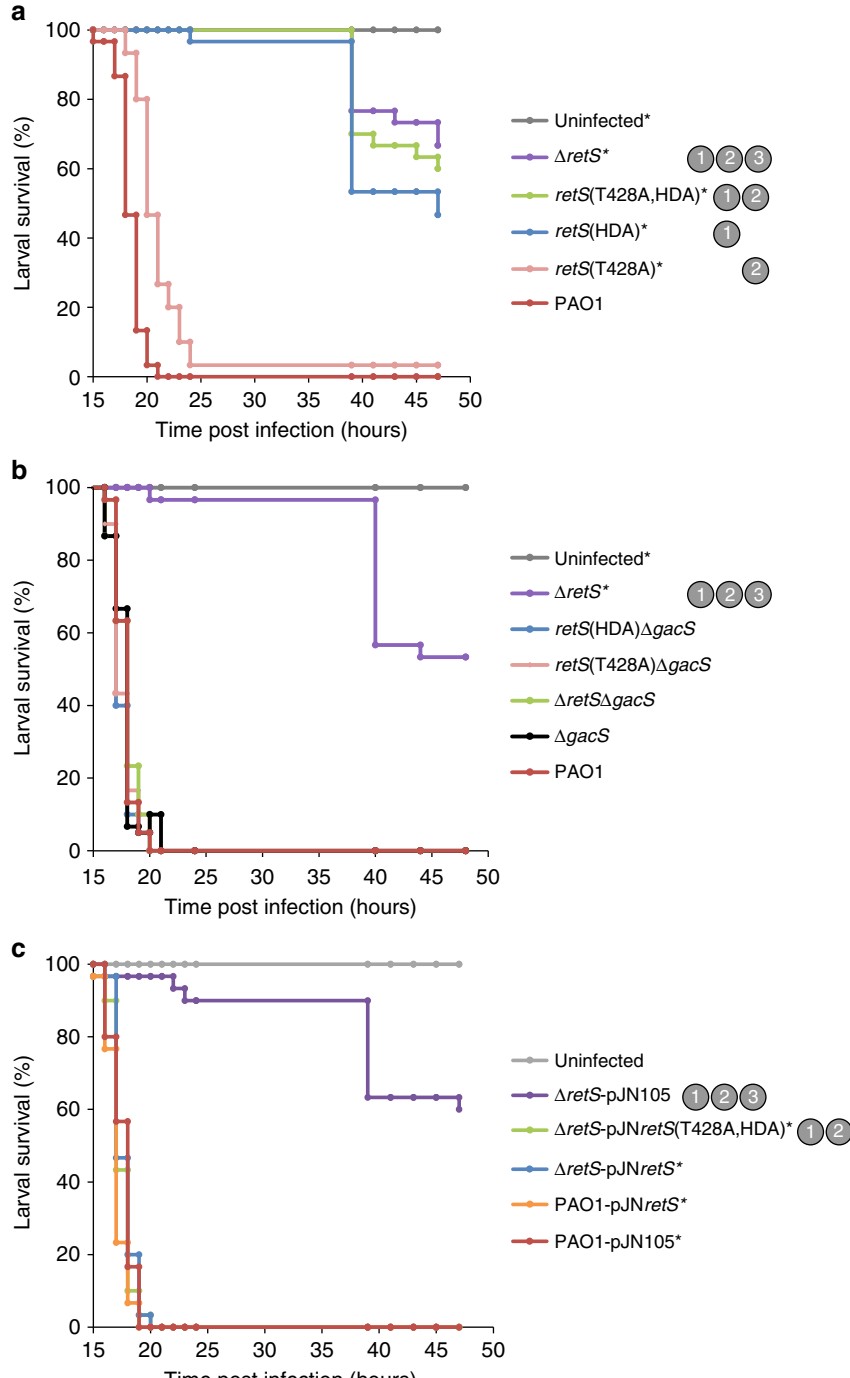

**Fig. 4** Mechanisms 1 and 2 are important for virulence in *Galleria mellonella*. **a** Survival of *G. mellonella* larvae infected with the *retS* mutant strains. *Indicates strains showing significantly attenuated virulence compared to PAO1 (*P* < 0.01, Mantel-Cox Log Rank with Bonferroni's correction for multiple comparisons). **b** The *retS* point mutations have no phenotype in a Δ*gacS* background. *Indicates strains showing significantly attenuated virulence compared to PAO1 (*P* < 0.01, Mantel-Cox Log Rank with Bonferroni's correction for multiple comparisons). **c** Complementation of the Δ*retS* mutant by overexpressing RetS(T428A,HDA). *Significantly more virulent than the Δ*retS*-pJN105 strain (*P* < 0.01, Mantel-Cox Log Rank with Bonferroni's correction for multiple comparisons). **a–c** The mechanisms absent from each strain are indicated in grey circles (except in the Δ*gacS* background strains that lack the target of these mechanisms). Three independent experiments were conducted each with ten larvae per strain (30 larvae in total per mutant). The inoculum was 20–40 CFUs per larvae

mice infected with the Δ*retS* mutant or the mutant lacking mechanism 1 (*retS*(HDA)) survived for the 7-day duration of the experiment, demonstrating that RetS and mechanism 1 are essential for virulence (Fig. 5). The *retS*(T428A) mutant, which

lacks mechanism 2, was also severely attenuated in virulence with 80% of mice surviving to the endpoint of the experiment (Fig. 5). These results indicate that mechanisms 1 and 2 are both important for acute virulence.

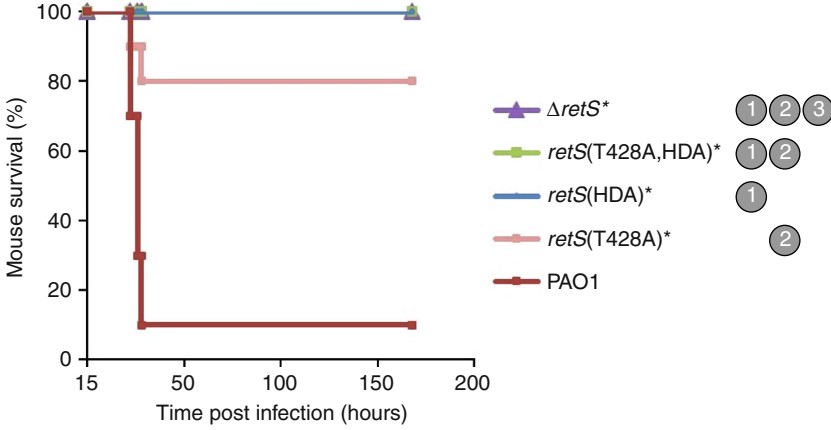

**Fig. 5** Mechanisms 1 and 2 are important for virulence in mice. Survival of mice infected intranasally with the *retS* mutant strains. Mice infected with the wild-type PAO1 strain developed an acute respiratory infection. Two independent experiments were performed, each with five mice per mutant strain (ten mice per mutant strain). The inoculum was $2 \times 10^7$ CFUs per mouse. *Indicates strains showing significantly attenuated virulence compared to PAO1 ($P < 0.01$, Mantel-Cox Log Rank with Bonferroni's correction for multiple comparisons). The mechanisms absent from each strain are indicated in grey circles

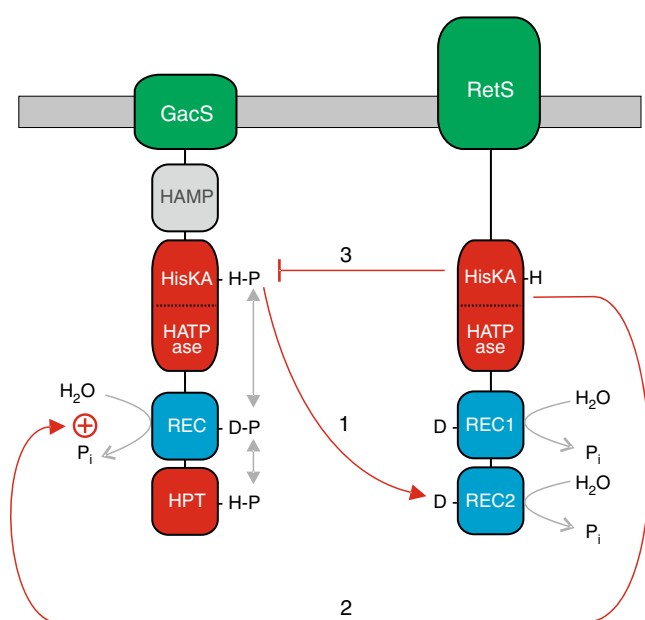

**Fig. 6** The three mechanisms used by RetS to inhibit GacS signalling. Mechanism 1—D858 in REC2 of RetS takes the phosphoryl group from the autophosphorylatable His residue in the HisKA domain of GacS. Mechanism 2—RetS uses its transmitter phosphatase activity to accelerate the dephosphorylation of the REC domain of GacS-P. Mechanism 3—The catalytic core of RetS inhibits the GacS autophosphorylation reaction

## Discussion

We have shown that two SKs, RetS and GacS, which play a major role in controlling virulence in *P. aeruginosa*, interact extensively, with RetS having three distinct mechanisms for downregulating GacS signalling (Fig. 6). In mechanism 1, RetS takes phosphoryl groups from GacS-P, with phosphotransfer occurring from the autophosphorylation site of GacS to REC2 of RetS. In mechanism 2, RetS has transmitter phosphatase activity that speeds up the dephosphorylation of the REC domain of GacS-P. In mechanism 3, RetS inhibits the autophosphorylation of GacS. These three mechanisms allow RetS to have exquisite control of GacS signalling and they all play significant roles in vivo and during infection. In particular, we show that mechanisms 1 and 2 play vital roles during acute respiratory infection in mice. This is the

first discovery of such an intricate level of interconnectedness and communication between a pair of SKs in a sensory network. This has profound implications for other sensory networks employing multiple SKs which, in light of this study, could be expected to use similar mechanisms of communication to process sensory data and integrate multiple signals. A further implication is that because of the increased number of different mechanisms by which kinase-to-kinase signalling can occur, multikinase networks may be even more widespread than previously appreciated.

The three mechanisms identified here represent an unprecedented level of communication between two SKs and there is great potential for them to be employed in other multikinase networks. Mechanism 1 employs intermolecular phosphotransfer to signal from the HisKA domain of one SK (GacS) to a REC domain located within another SK (RetS). Here, REC2 of RetS functions as a phosphate sink for GacS. As around 20% of SKs have a REC domain (i.e. are hybrid/unorthodox), related phosphotransfer mechanisms will be found connecting many other pairs of kinases that work together[12,13,18,46]. Mechanism 2 uses the transmitter phosphatase activity of one SK (RetS) against the REC domain of another SK (GacS); as far as we are aware, this is the first demonstration of transmitter phosphatase activity occurring between two kinases, but there is wide potential for it to be employed in other multikinase networks because almost all SKs have transmitter phosphatase activity[20]. Mechanism 3 involves inhibitory interactions between the catalytic cores of two kinases (RetS and GacS)[10], and consistent with this mechanism being used by other networks are the findings of a systematic two-hybrid screen of the SKs from *Myxococcus xanthus*, which tested 725 possible catalytic core/catalytic core interactions and found evidence of interaction in over 100 cases[16].

Mechanism 1 depends on the phosphorylatable aspartate residue in REC2 of RetS (D858) and is essential for virulence, biofilm formation, swarming and normal expression levels of RsmY&Z. Most strikingly, the *retS*(HDA) mutant, lacking D858, was completely avirulent in mice (Fig. 5). Prior to our discovery of its key role in mechanism 1, the role of D858 had been investigated. Similar to our findings with the PAO1 strain of *P. aeruginosa*, Laskowski and Kazmierczak found that for the PA103 strain, D858 is essential for RetS function[56]. However, in contrast, Goodman et al. found no phenotype for the D858 mutation in the PAK strain[10]. This suggests that either of the two remaining mechanisms (which are independent of D858)

compensate when mechanism 1 is lost in the PAK strain but not in the PAO1 and PA103 strains.

The three mechanisms that we have demonstrated signal via GacS to control phosphorylation levels of the output RR, GacA, which controls the expression of the regulatory RNAs, RsmY and RsmZ. Loss of any individual mechanism is sufficient to increase RsmY expression levels to the same extent as seen in the ΔretS mutant (Fig. 2b), whereas although significant elevation in RsmZ levels is seen when individual mechanisms are lost, it is not to the same extent as seen with the ΔretS mutant (Fig. 2c). RsmY expression has previously been reported to be at least 2-fold higher than RsmZ expression[55,57]. These findings can be explained by a model where GacA-P binds more tightly to the RsmY promoter than to the RsmZ promoter. Deleting retS, eliminates all three of our mechanisms, thereby giving a large rise in GacA-P levels, sufficient to fully activate RsmY and RsmZ expression. The retS point mutations, by disabling individual mechanisms, would generate a rise in GacA-P levels but not as much as is seen in the retS deletion mutant. This lesser rise in GacA-P levels would be enough to fully activate the RsmY promoter (due to its higher binding affinity) but not enough to fully activate the RsmZ promoter.

An intriguing question is why are multiple mechanisms necessary to orchestrate this virulence switch? Presumably, each mechanism contributes uniquely to the balance of the decision-making process during infection and the complexity of this process reflects the importance of the decision to bacterial survival in the host. Differential regulation of the three mechanisms would allow precise control of GacS signalling and, given their importance for virulence, it is tempting to speculate that these elaborate mechanisms constitute a logic gate for processing and integrating the different stimuli sensed by GacS and RetS to decide the course of the infection. Although all three mechanisms allow RetS to downregulate GacS signalling, there are significant differences. For example, mechanism 3 blocks the autophosphorylation of GacS but it differs from mechanisms 1 and 2, as it is unable to dephosphorylate GacS-P. This is important because autophosphorylation is not the only way of generating GacS-P, as LadS-P phosphorylates GacS[46]; GacS-P generated from LadS-P would be unaffected by mechanism 3 but could be targeted by mechanisms 1 and 2.

How might the three different mechanisms be controlled? RetS has a periplasmic ligand binding domain that has been implicated in detecting kin-cell lysis but its ligand is currently unknown[50,58,59]. Ligand binding could regulate any of the three mechanisms, but in other SKs there is a strong precedent for it regulating the balance between kinase activity and transmitter phosphatase activity[20]. RetS lacks kinase activity but, following this precedent, transmitter phosphatase activity (mechanism 2) is very likely to be under ligand control. Mechanism 1 provides considerable potential for linkage to other signalling pathways, in particular the HptB signalling pathway. HptB is a single domain Hpt protein that relays phosphoryl groups from several hybrid kinases (PA1611, ErcS' and SagS) to the output RR, HsbR, which indirectly controls motility and cyclic-di-GMP levels[54,60,61]. Analogous to how RetS serves as a phosphoacceptor for GacS-P in mechanism 1, RetS can also take phosphoryl groups from HptB-P[60]. This could provide a route for HptB signalling to downregulate mechanism 1, since when RetS is phosphorylated by HptB-P, then, until it has dephosphorylated, it will be unable to accept phosphoryl groups from GacS-P. This potential communication route would expand the number of SKs, and therefore the number of different signals, that could influence signalling by the GacS network. Aside from this HptB-mediated link, PA1611 has been shown to interact directly with RetS[51,52]. In wild-type cells, PA1611 is expressed only at very low levels making the physiological relevance of this interaction uncertain[51]; however,

it has been found that overexpression of PA1611 using a multicopy plasmid expression vector favours the interaction between PA1611 and RetS, and relieves GacS from the inhibitory effects of RetS, suggesting that PA1611 sequesters RetS away from GacS[51]. The phenotypic data from PA1611 overexpression are consistent with a total loss of RetS function[51,52], suggesting that the sequestration of RetS by PA1611 blocks all three of the mechanisms by which RetS targets GacS.

In conclusion, we have discovered extensive interactions between the GacS and RetS SKs that play a critical role in controlling the switch between acute and chronic infection. We have identified three distinct biochemical mechanisms and demonstrated their important roles in vivo and in insect and mouse infection models. The complexity of these mechanisms reflects the importance of the finely balanced decisions that the GacS network makes during infection. As these mechanisms involve highly conserved domains or sequence motifs, they are likely to be used by many other multikinase networks for signal integration and decision-making.

## Methods

**Bacterial strains and growth conditions**. Bacterial strains and plasmids are described in Supplementary Table 1. Unless otherwise stated, bacteria were grown in LB broth at 37 °C. When used, M63 (2 g/l (NH₄)₂SO₄, 13.6 g/l KH₂PO₄ and 0.5 mg/l FeSO₄) was supplemented with 1 mM MgSO₄, 0.5% casamino acids and 0.2% glucose. Antibiotics were used at the following concentrations: ampicillin 100 µg/ml, kanamycin 25 µg/ml, tetracycline 50 µg/ml and gentamycin 25 µg/ml (Escherichia coli) or 100 µg/ml (P. aeruginosa).

**Plasmid construction**. Genes for overexpressing wild-type cytoplasmic portions of proteins were amplified from P. aeruginosa PAO1 genomic DNA using primers described in Supplementary Table 2. Point mutations were introduced using overlap extension PCR. Wild-type genes and their mutant derivatives were cloned into the pQE60 expression plasmid which attaches a C-terminal 6xHis tag. The proteins were overexpressed and purified as previously described for other SKs[62,63]. Allelic exchange plasmids for in-frame deletion of genes in PAO1 or for introducing point mutations were constructed using primers described in Supplementary Table 2 using P. aeruginosa DNA as a template. These constructs were cloned into pEX19Gm for allelic exchange with PAO1[64,65].

**Strain construction**. In-frame gene deletions and allelic exchange of gene regions containing introduced point mutations were carried out by tri-parental mating using E. coli containing the mobilisation plasmid, pRK2013[66]. Subsequent sucrose and gentamycin susceptibility tests were done to isolate potential mutants. Deletion mutants were checked via PCR using primers outside of the initial construct used to make the deletion. PCR products were sequenced to confirm mutations. Tetra-primer PCR[67] was used as a preliminary screen to identify strains containing desired point mutations. Potential mutants were then checked by sequencing using PCR products obtained using primers outside of the original mutation construct.

**Autophosphorylation assays**. Reactions were performed in TGMNKD buffer (10% (v/v) glycerol, 150 mM NaCl, 50 mM Tris HCl, 1 mM DTT, 5 mM MgCl₂, 50 mM KCl, pH 8.0) and initiated by the addition of 2 mM [γ³²P] ATP (3.7 GBq/mmol PerkinElmer). The reactions contained 5 µM GacS derivative and 20 µM RetS derivative (Fig. 1m, n). The final reaction volume was 100 µl; 10 µl aliquots were taken at the indicated timepoints and quenched in 20 µl of 2× SDS loading dye (7.5% (w/v) SDS, 90 mM EDTA, 37.5 mM Tris pH 6.8, 37.5% glycerol, and 3% β-mercaptoethanol). Samples were stored on ice and then analysed using SDS-PAGE (10% (w/v) polyacrylamide). Gels were exposed to phosphorscreens (Fuji) for 1 h and then analysed using a Fujifilm FLA-7000 phosphorimager. The uncropped phosphorimages used to produce Fig. 1 and Supplementary Fig. 1 are shown in Supplementary Figs. 5 and 6.

**Pre-phosphorylation of GacS derivatives**. GacSc, GacSc(HAQ) and GacSc (HDQ) were purified and then incubated with 2 mM [γ³²P] ATP (3.7 GBq/mmol PerkinElmer) for 1 h at 20 °C. The phosphorylated proteins were then diluted in lysis buffer and purified away from unincorporated ATP using a Ni-NTA column[68,69].

**GacS-P dephosphorylation assays**. Reaction tubes contained TGMNKD buffer and, where appropriate, RetSc or one of its mutant derivatives. The reactions were initiated by addition of phosphorylated GacS (either wild-type GacSc-P or one of its mutant derivatives). Reactions contained: 2 µM GacSc-P and 50 µM RetSc derivative (Fig. 1a–d and Supplementary Fig. 1a–c), 4 µM GacSc(HAQ)-P and

20 μM RetSc derivative (Fig. 1e–h and Supplementary Fig. 1d–f), 2 μM GacSc-P and 20 μM RetSc derivative (Fig. 1j, k), and 2 μM GacSc(HDQ)-P and 20 μM RetSc derivative (Supplementary Fig. 2). Reactions were performed at 20 °C; 10 μl samples were taken at the timepoints indicated and processed as described for the autophosphorylation assays. Half-times of GacSc-P dephosphorylation were calculated using Origin 4.1. Data were analysed using a one-way ANOVA with Tukey's modification.

**Biofilm formation.** Biofilm formation was measured using the MBEC™ (Minimum Biofilm Eradication Concentration) Assay from Innovotech. This features a 96-well plate with a peg lid and was used according to a modified method previously described[70]. Overnight LB cultures were standardised in LB broth to an $OD_{600nm} = 1.0$, which were then diluted 1:100 in M63. In each well, 150 μl of diluted culture or uninoculated broth was dispensed before the sterile peg lid was sealed on the plate. Each strain had five technical repeats per plate. The plates were incubated for 10 h at 37 °C with shaking at 125 rpm. Peg lids were removed and washed in PBS before being dried at 65 °C. Dried lids were stained with 0.1% (w/v) crystal violet. The pegs were washed three times in PBS, 5 min per wash, before bound crystal violet was solubilised in 95% ethanol. These plates were read at $OD_{570nm}$. Three biological repeats were performed.

**RNA extraction, cDNA and qPCR.** Overnight cultures were subcultured into LB broth with a starting $OD_{600nm}$ of 0.03 and incubated for 6 h at 37 °C with shaking. Cells were harvested and processed following the supplier's protocol through the RiboPure Bacteria Kit (Ambion), which includes a DNA removal step, which was repeated twice. Purified RNA was checked by PCR for DNA contamination before cDNA was made using SuperScript III reverse Transcriptase (Life Technologies) following the supplier's protocol. Quantitative real time PCR (qRT-PCR) was carried out on a Stratagene Mx3005P machine using Brilliant III Ultra-Fast SYBR Green qPCR kit (Agilent Technologies) following the manufacturer's protocol. The oligonucleotides used are detailed in Supplementary Table 2. The housekeeping gene *rpoC* was used an internal control. Fold changes in expression were calculated using the $2^{-\Delta\Delta C_T}$ method[71]. Data were analysed using a one-way ANOVA with contrasts.

**Swarming motility assay.** Large swarming plates (140 mm) contained 0.5% (w/v) agar (LAB Agar No.2 Bacteriological) and 8 g/l nutrient broth (Oxoid) supplemented with 0.5% (w/v) glucose. Plates were inoculated with 0.5 μl overnight culture, incubated at 20 °C for 2 h prior to incubation at 30 °C for 16 h. Plates were then incubated at 20 °C for a further 8 h before imaging.

**Galleria mellonella infection model.** *Galleria* larvae killing was performed based on a previously described assay[72]. *Galleria* larvae were obtained from UK Waxworms Ltd. (Sheffield, UK). *P. aeruginosa* stains were grown overnight in M63 before being centrifuged, washed and resuspended in PBS to an $OD_{590} = 1\pm0.05$. This was diluted $5 \times 10^5$ fold in PBS. Using a gastight repeat dispenser Hamilton syringe, $2 \times 5$ μl of the diluted bacterial suspension (20–40 CFUs) were injected into the hindmost proleg of the larvae. Larvae were incubated at 37 °C and their survival was monitored after 15 h of incubation for up to 47 h. Data were analysed using Kaplan–Meier survival curves. Statistical significance was assessed using the Mantel-Cox Log Rank test, applying Bonferroni's correction for multiple comparisons.

**Mouse respiratory infection model.** The mouse infection model was done as previously described[73]; 7–9-week-old female Balb/c mice (Harlan) were anaesthetised with $O_2$ and isoflurane and infected intranasally with a challenge dose of $2 \times 10^7$ CFUs. Two independent experiments were performed, each with five mice per strain. Power calculations were used to estimate the number of mice needed to detect a 10% change in the mean survival time. Mice were randomly assigned to each of the treatment groups. No blinding was used. Survival was followed over 7 days and scored as previously described[73]. Mouse experiments were approved by the United Kingdom Home Office (Home Office Project License Number 40/3602) and the University of Liverpool Animal Welfare and Ethics Committee.

**Western blot.** Overnight LB cultures were standardised in LB broth to an $OD_{600nm} = 1.0$, which were then diluted 1:100 in M63. Cells were incubated for 10 h at 37 °C with shaking at 125 rpm; 1 ml of cells was centrifuged at 13,000 rpm. Cell pellets were resuspended in 100 μl of BugBuster (Merck) and incubated at 20 °C for 20 min to lyse the cells; 25 μl NuPAGE® LDS sample buffer (4X) (Invitrogen) was added and samples were boiled for 5 min at 100 °C. Whole cell lysates were separated using Bis-Tris NuPAGE® Novex® 4–12% SDS-PAGE gels (Invitrogen). Proteins were electrically transferred to a nitrocellulose membrane (Fisher) and blocked overnight at 4 °C using 3% (w/v) skimmed milk (Sigma) in TPBS (PBS supplemented with 1% (v/v) Tween-20). The membrane was incubated at 20 °C for 30 min before being incubated for 90 min with 3% (w/v) skimmed milk in TPBS with the primary antibody (rabbit) diluted 1:1000. The primary rabbit antibody was raised against the purified cytoplasmic portion of RetS (Eurogentec). The membrane was washed three times for 5 min in TPBS. It was then incubated for 90 min

with a secondary antibody (goat anti-rabbit) with an infrared dye (IRDye® 800 W secondary antibodies from *Li-Cor*) diluted 1:25000 in TPBS with 3% (w/v) skimmed milk. The membrane was washed three times for 5 min with TPBS and imaged on the *Li-Cor* Odyssey.

**Data availability**. The authors declare that the data supporting the findings of this study are available within the paper and its supplementary information files.

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

## Acknowledgements

This work was supported by the Medical Research Council (MRC) (grant number MR/M020045/1), the Leverhulme Trust (grant number RPG-2014-228), the RoseTrees Trust (grant number M328) and a NERC PhD studentship (grant number 1076449). We are grateful to Lily Byham, Charlotte Unwin and Adam Wise for their assistance. We would like to thank Alain Filloux for useful discussions.

## Author contributions

V.I.F., S.E.F.-J., A.G., E.M.W. and S.L.P. conducted the experiments; V.I.F., E.M.W., A.K., A.R.B. and S.L.P. designed the experiments and wrote the paper.

## Additional information

**Competing interests:** The authors declare no competing interests.

