## [Peer Review File · Nature Communications]

Reviewers' comments:

Reviewer #1 (Remarks to the Author):

The manuscript "Multiple communication mechanisms between sensor kinases are critical for virulence in *Pseudomonas aeruginosa*" by Francis et al describes the identification of three distinct mechanisms of interaction between RetS and GacS. In bacteria, complex interactions among multiple sensor kinases (SKs) and response regulators (RRs) are common and play important role for connecting environmental signals to cellular processes. Identification of detailed mechanisms of the interaction between RetS and GacS in *P. aeruginosa* is not only of interest to *Pseudomonas* research field but potential to the broad field of bacteriology. However, a few concerns dampened the enthusiasm of the reviewer.

1. Since the only output of the interaction between RetS and GacS is the phosphorylation status of GacS, it is difficult to see the importance of the different interaction mechanisms in the cell. No evidence is presented that indicate the different mechanisms respond to different signals.

2. I'm not sure that I'm completely convinced by the data that support mechanism 1. In figure 1f, the decrease of GacSc(HAQ)-P is considered an evidence that phosphotransfer had occurred from GacSc(HAQ)-P to RetSc, then, why RetSc-P bands did not increased? It is noted that in this case auto-dephosphorelatio was not present in GacS(HAQ)-P. Also, while I understand the difficulty of these experiments, the images (1a to 1h) are of quite low quality and not to the standard of Nat Comm, probably not to any other good journals.

3. When study the effect of different point mutations, e.g. D858, no control mutations were used. It is possible that these point mutations had general structural effects, instead of the particular amino acid residue, which cause the changes observed. It would be helpful to include a mutant that has point mutation near the targeted residue.

Minor issue: Page 20, line 7, delete "this"

Reviewer #2 (Remarks to the Author):

Francis et al present a manuscript titled "Multiple communication mechanisms between sensor kinases are critical for virulence in *Pseudomonas aeruginosa* ». This manuscript describes 3 roles of the RetS histidine kinase by which this sensor attenuates GacS signalling. The authors study impact of the recombinant RetS proteins on GacS phosphorylation state but also impacts on these retS mutants on biofilm control, rsmY and rsmZ expression and on several animal models. At the ends the authors present an unprecedented level of cross-communication between GacS and RetS sensors. The manuscript makes an important and novel contribution to our understanding of the role of RetS in *P. aeruginosa*.

Major points and suggestions.

- The authors show that in mechanism 1, RetS takes phosphoryl groups from GacS-P by using its phosphorylation site in REC2 (D858) [RetSc(HDA)]. They suggest that the Rec1 domain is not involved in this mechanism since the RetSc(HAA) protein, where Rec1 and Rec2 are inactivated, exhibits similar behavior as the RetSc(HDA). To conclude that Rec1 doesn't playing any role in mechanism 1 the most appropriate controls is to use a RetSc(HAD) mutant where the conserved aspartate in Rec1 is

mutated.

- In mechanism 3 the authors show that it requires only the HK domain of GacS and RetS proteins. To prove this, they show that the presence of RetSc(HAA) in a GacSc(HAQ) autophosphorylation reaction, reduces the level of GacSc(HAQ)-P. A similar result is observed when using only the kinase catalytic cores (HisKA and HATPase domain) of GacS and RetS. However, in these both cases the mechanism 2 depending of the T428A in the HisKA domain is still there. To avoid any possible implication on the RetS phosphatase activity, it could be interesting to see the effect of RetSc (T428A, HDA) on GacSc (HAQ) autophosphorylation reaction.

- How can explain that RsmY and RsmZ expression (that are both under the direct control of GacA) present a so different behavior in response to the 3 RetS mechanisms? There is any evidence that GacA-P have different affinity for RsmY or RsmZ promoters?

- This manuscript is focused on the 3 roles of the RetS histidine kinase, however there is not mention about the regulation of these mechanisms. Kong and al 2013 shows that RetS action could be counteracted by the PA1611. Can there be a role of PA1611 on the 3 mechanisms? In other word what's happen if PA1611 is overexpressed when RetS is added in experiment figure 1F, 1M and 1K?

- The discussion is focused on the link between RetS and GacS sensors. In 2009 Hsu et al show that RetS also Interacts with HptB protein and they propose that RetS could dephosphorylates this protein. Thus, the new mechanisms presented in this work could be the missing link between these two pathways. perhaps the authors would add a comment on that to open the discussion.

In a previous work Goodman et al., found no phenotype for the D858 mutation in the PAK strain. The authors propose that this difference could be the fact that RetS is more highly expressed in the PAK strain than in the PAO1 or PA103. I find this sentence a little bit weird, there is no evidence that RetS has a higher expression in the PAK strain than in the PAO1 or PA103. Are there any differences in the promoter sequences explaining a putative difference in RetS regulation?

We would like to thank the reviewers for their careful reading of our manuscript and for their constructive comments. We have addressed all of their comments below and in the revised manuscript.

Reviewer #1 (Remarks to the Author):

The manuscript “Multiple communication mechanisms between sensor kinases are critical for virulence in Pseudomonas aeruginosa” by Francis et al describes the identification of three distinct mechanisms of interaction between RetS and GacS. In bacteria, complex interactions among multiple sensor kinases (SKs) and response regulators (RRs) are common and play important role for connecting environmental signals to cellular processes. Identification of detailed mechanisms of the interaction between RetS and GacS in P. aeruginosa is not only of interest to Pseudomonas research field but potential to the broad field of bacteriology. However, a few concerns dampened the enthusiasm of the reviewer.

1. Since the only output of the interaction between RetS and GacS is the phosphorylation status of GacS, it is difficult to see the importance of the different interaction mechanisms in the cell. No evidence is presented that indicate the different mechanisms respond to different signals.

The only output of the GacS/RetS interaction is the level of GacA-P in the cell. All three of the mechanisms that we have uncovered will contribute to setting this level and this will determine the extent to which the two regulatory RNAs, RsmY and RsmZ, are expressed and therefore the activity of RsmA. RsmA affects the expression of over 500 genes¹, and its activity is titrated by RsmY and RsmZ. When GacS is off, levels of RsmY and RsmZ would be very low, and therefore RsmA activity maximal. In contrast, very high GacS activity would lead to high GacA-P levels (as would be seen in $\Delta retS$ strain) and would be sufficient to fully activate RsmY and RsmZ expression, completely deactivating RsmA. More moderately elevated levels of GacA-P (as would be seen in strains lacking an individual mechanism) give more graded responses, with high-level RsmY expression but comparatively lower levels of RsmZ expression (Fig. 2b,c; see our response to Reviewer 2 on the differential effects of the three mechanisms on RsmY/Z expression), and this would deliver a more intermediate level of RsmA activity that would affect some of the downstream targets more than others, depending on their binding affinity for RsmA. Therefore, there will be a spectrum of different responses, rather than a simple on/off switch, according to how much GacS activity is present, with the three mechanisms that we have uncovered making a major contribution to setting this level and therefore to the cellular response.

We have reworked the discussion, to highlight that the mechanisms are not equivalent. A key difference stems from there being two different ways of generating GacS-P; either by GacS autophosphorylation or by phosphotransfer from LadS². Mechanism 3 blocks GacS autophosphorylation but it can not prevent GacS from being phosphorylated by LadS. In contrast, as mechanisms 1 and 2 dephosphorylate GacS-P, they can counteract both GacS autophosphorylation and phosphotransfer from LadS. We have also added a paragraph to the discussion section exploring the different ways that the mechanisms could be regulated. This includes: i) a discussion of the role of an unidentified signalling molecule that allows detection of kin-cell lysis and possible links to mechanism 2, ii) the role of HptB to RetS phosphotransfer and how it could affect mechanism 1 and iii) the PA1611/RetS interaction as a means of regulating all three mechanisms.

2. I'm not sure that I'm completely convinced by the data that support mechanism 1. In figure 1f, the decrease of GacSc(HAQ)-P is considered an evidence that phosphotransfer had occurred from GacSc(HAQ)-P to RetSc, then, why RetSc-P bands did not increased? It is noted that in this case autodephosphorelation was not present in GacS(HAQ)-P. Also, while I understand the difficulty of these experiments, the images (1a to 1h) are of quite low quality and not to the standard of Nat Comm, probably not to any other good journals.

All proteins containing functional receiver domains have the ability to autodephosphorylate^{3,4}. For this reason, GacSc-P and RetSc-P are both capable of autodephosphorylation but GacSc(HAQ)-P is not. In Figure 1f, the reason that GacSc(HAQ)-P levels fell over time is due to phosphotransfer to RetSc. The reason why RetSc-P levels did not rise continually throughout this phosphotransfer reaction is because RetSc-P autodephosphorylates rapidly, so that phosphoryl groups are lost as quickly from RetSc as they are being supplied from GacSc(HAQ)-P. We have added a sentence to the manuscript explaining this.

The image quality of Fig1a-h is a consequence of the rapid dephosphorylation of RetS-P, which means RetSc-P only accumulates to fairly low levels and we are working close to our detection limit. We have replaced these images with clearer higher resolution images and also included supplementary figures (Supplementary Fig. 5 & 6) showing the entire gels from which these images were sourced.

3. When study the effect of different point mutations, e.g. D858, no control mutations were used. It is possible that these point mutations had general structural effects, instead of the particular amino acid residue, which cause the changes observed. It would be helpful to include a mutant that has point mutation near the targeted residue.

We used two *retS* mutations to study RetS function, the phosphorylation site in REC2 was disabled by the D858A substitution, while the T428A substitution (within the HExxT phosphatase motif), disabled HisKA transmitter phosphatase activity. The RetS(T428A), RetS(D858A) (aka RetSc(HDA)) and RetS(T428A,D858A) (aka RetSc(T428A,HDA)) mutant proteins purified as well as wild-type RetS from *E. coli* and we verified using Western blotting that they are expressed as well as wild-type RetS in *P. aeruginosa* (Supplementary Fig. 3). It is unlikely that a protein with a substitution causing general structural effects would be expressed to the same level as the wild-type protein in either *E. coli* or *P. aeruginosa*. In addition, proteins bearing the T428A and D858A mutations still retained biochemical activity (see Fig. 1k & m); for example, RetSc(T428A,HDA) contains the T428A and D858A substitutions and is therefore disabled for mechanisms 1 and 2 but retained the ability to perform mechanism 3 (Fig. 1m). As the mutant proteins retain biochemical activity they cannot be misfolded. Moreover, the substitutions that we have employed to study the role of the REC phosphorylation site (D858A) and the HisKA transmitter phosphatase activity (T428A) have a long history of being used to study these activities in other two-component systems without causing misfolding⁵⁻⁷. Furthermore, perhaps the most compelling evidence that these mutations do not cause generalised structural defects are the crystal structures of these substitution mutants from other kinases/response regulators; mutation of the phosphorylation site of the response regulator CheY (D57A, which corresponds to the D858A substitution in RetS) and mutation of the conserved threonine in the HExxT transmitter phosphatase motif in HK853 (T264A, which corresponds to T428A in RetS) does not cause structural changes^{8,9}. For these reasons we are confident that the site directed mutations that we have employed have not caused any wider structural issues.

Minor issue: Page 20, line 7, delete "this"

Done

Reviewer #2 (Remarks to the Author):

Francis et al present a manuscript titled "Multiple communication mechanisms between sensor kinases are critical for virulence in Pseudomonas aeruginosa ». This manuscript describes 3 roles of the RetS histidine kinase by which this sensor attenuates GacS signalling. The authors study impact of the recombinant RetS proteins on GacS phosphorylation state but also impacts on these retS mutants on biofilm control, rsmY and rsmZ expression and on several animal models. At the ends the authors present an unprecedented level of cross-communication between GacS and RetS sensors. The manuscript makes an important and novel contribution to our understanding of the role of RetS in P. aeruginosa.

Major points and suggestions.

- The authors show that in mechanism 1, RetS takes phosphoryl groups from GacS-P by using its phosphorylation site in REC2 (D858) [RetSc(HDA)]. They suggest that the Rec1 domain is not involved in this mechanism since the RetSc(HAA) protein, where Rec1 and Rec2 are inactivated, exhibits similar behavior as the RetSc(HDA). To conclude that Rec1 doesn't playing any role in mechanism 1 the most appropriate controls is to use a RetSc(HAD) mutant where the conserved aspartate in Rec1 is mutated.

This is a good point and we are grateful to the reviewer for requesting this. We have now included the RetSc(HAD) data as a supplementary figure (Supplementary Fig. 1). The rate of phosphotransfer from GacSc-P to wild-type RetSc and to RetSc(HAD) is similar, indicating that the conserved aspartate in REC1 plays no role in mechanism 1.

- In mechanism 3 the authors show that its requires only the HK domain of GacS and RetS proteins. To prove this, they show that the presence of RetSc(HAA) in a GacSc(HAQ) autophosphorylation reaction, reduces the level of GacSc(HAQ)-P. A similar result is observed when using only the kinase catalytic cores (HisKA and HATPase domain) of GacS and RetS. However, in these both cases the mechanism 2 depending of the T428A in the HisKA domain is still there. To avoid any possible implication on the RetS phosphatase activity, it could be interesting to see the effect of RetSc (T428A, HDA) on GacSc (HAQ) autophosphorylation reaction.

Mechanism 2 requires T428 in the HisKA domain of RetS and targets the aspartate residue in the REC domain of GacS. As GacSc(HAQ) and GacSHK lack the aspartate residue that is targeted by mechanism 2, there is no possibility that mechanism 2 could be operating in any of the experiments shown in Fig. 1m & n. That said, we agree with the reviewer that it would be a very useful control to show the data demonstrating the effect of RetSc(T428A,HDA) on GacS(HAQ) autophosphorylation. We have now included this in Fig. 1m (bottom panel). Just like RetSc(HAA), RetSc(T428A,HDA) can inhibit GacS autophosphorylation (Fig. 1m), therefore we can conclude that T428 is not involved in mechanism 3.

- How can explain that RsmY and RsmZ expression (that are both under the direct control of GacA) present a so different behavior in response to the 3 RetS mechanisms? There is any evidence that GacA-P have different affinity for RsmY or RsmZ promoters?

We have added an explanation to the discussion. RsmY and RsmZ are known to have very different promoter structures; both include binding sites for GacA but RsmY expression appears to be around 2-fold higher than that of RsmZ^{10,11}. Our results are consistent with a model where GacA-P binds more tightly to the RsmY promoter than to the RsmZ promoter. Deleting *retS*, eliminates all three of our mechanisms, thereby giving a large rise in GacA-P levels sufficient to fully activate RsmY and RsmZ expression. The *retS* point mutations, by disabling individual mechanisms, would generate a rise in GacA-P levels but not as much as is seen in the *retS* deletion mutant. This lesser rise in GacA-P levels would be enough to fully activate the RsmY promoter (due to its higher binding affinity) but not enough to fully activate the RsmZ promoter.

- This manuscript is focused on the 3 roles of the RetS histidine kinase, however there is not mention about the regulation of these mechanisms. Kong and al 2013 shows that RetS action could be counteracted by the PA1611. Can there be a role of PA1611 on the 3 mechanisms? In other word what's happen if PA1611 is overexpressed when RetS is added in experiment figure 1F, 1M and 1K?

We have expanded the discussion to include potential regulatory routes. Phosphorylation of RetS by HptB being one route that could regulate Mechanism 1. As the reviewer states, PA1611 has been shown to interact with RetS¹², but its physiological role is unclear and its role in virulence is untested. The phenotype resulting from deletion of *pa1611* is extremely mild having no effect on either swarming motility or cytotoxicity against mammalian cells in culture¹². A small increase in ExoS (a T3SS toxin) promoter activity was noted in the $\Delta pa1611$ mutant but as this did not affect cytotoxicity (which depends on T3SS) its importance is unclear¹². A slight decrease in biofilm formation was seen for the $\Delta pa1611$ mutant¹³. Strong phenotypes for PA1611 are only seen when PA1611 is overexpressed using expression plasmids. Like the $\Delta retS$ mutant, strains overexpressing PA1611 have increased biofilm formation, increased expression of RsmY & RsmZ, decreased swarming, decreased ExoS expression and decreased cytotoxicity¹². This is consistent with PA1611 forming a complex with RetS that sequesters it away from GacS as originally proposed in the Kong et al. study¹², thus blocking all three of the mechanisms that we have discovered. However, at physiological expression levels of PA1611, the extent to which this complex forms appears to be small because deletion of *pa1611* only gives mild phenotypes.

Transcriptomics studies in mice, have shown that during acute and chronic infection, PA1611 is downregulated approximately fourfold compared to its expression levels in laboratory growth media¹⁴. This suggests that PA1611 plays, at most, a very minor role during infection. Consistent with this idea, PA1611 was not found to be an essential gene for either acute or chronic infection in Tn-Seq experiments performed in mice¹⁴. These data are consistent with PA1611 having little relevance during infection, and so we have not biochemically defined how PA1611 would affect the interactions between RetS and GacS. Outside of infection, if physiological conditions exist where PA1611 is highly expressed, then it would be entirely in keeping with all of the published phenotypic data on PA1611 overexpression, if PA1611 were to disrupt all three of our mechanisms by binding and sequestering RetS away from GacS. We have added this to the discussion section.

- The discussion is focused on the link between RetS and GacS sensors. In 2009 Hsu et al show that RetS also Interacts with HptB protein and they propose that RetS could dephosphorylates this protein. Thus, the new mechanisms presented in this work could be the missing link between these two pathways. perhaps the authors would add a comment on that to open the discussion.

We have expanded the discussion section as suggested. The *retS* mutant phenotypes that we have seen are completely dependent on signalling through GacS. We have introduced our *retS* point mutations into a $\Delta gacS$ background and found that these strains behave identically to their $\Delta gacS$ parent (Supplementary Fig. 4 and we have added new *Galleria mellonella* data in Fig. 5b). As wild-type *gacS* is required for our *retS* point mutations to give their phenotypes, this eliminates the possibility of the phenotypes of the *retS* point mutations being caused by some other aspect of GacS network signalling such as HptB signalling rather than GacS. However, as the reviewer suggests, there could still be a way for HptB signalling to influence RetS signalling to GacS. HptB-P has been shown to phosphorylate RetS¹⁵, and it is therefore plausible that this could be a way of regulating mechanism 1. Mechanism 1 depends on RetS accepting phosphoryl groups from GacS-P, but under conditions where HptB signalling is active and therefore RetS is being phosphorylated by HptB-P, then this could limit the ability of RetS to accept phosphoryl groups from GacS-P.

In a previous work Goodman et al., found no phenotype for the D858 mutation in the PAK strain. The authors propose that this difference could be the fact that RetS is more highly expressed in the PAK strain than in the PAO1 or PA103. I find this sentence a little bit weird, there is no evidence that RetS has a higher expression in the PAK strain than in the PAO1 or PA103. Are there any differences in the promoter sequences explaining a putative difference in RetS regulation?

Here we are attempting to reconcile the results of different studies on the phenotype caused by mutating D858 in different *P. aeruginosa* strains. A difference in the expression level of RetS between the strains would be a plausible explanation, given that we have shown that RetS expression levels affects whether D858 is required (at physiological expression levels D858 is required but when RetS is overexpressed D858 is not required).

While there are no differences in the *retS* promoter sequences, within the coding sequence of *retS* there are two consecutive rare threonine codons found in the PAO1 and PA103 strains (ACTACT; codons 42 and 43), which are both replaced with the much more commonly used ACC codon for threonine in the PAK strain. In *P. aeruginosa*, the ACC codon for threonine is used 19 times more frequently than ACT. This more favourable codon usage could facilitate translation in the PAK strain relative to the PAO1 and PA103 strains, allowing more RetS to be produced in the PAK strain. We have expanded our discussion to include this.

References

1. Brencic, A. & Lory, S. Determination of the regulon and identification of novel mRNA targets of *Pseudomonas aeruginosa* RsmA. *Mol. Microbiol.* **72**, 612-632 (2009).
2. Chambonnier, G. et al. The hybrid histidine kinase LadS Forms a multicomponent signal transduction system with the GacS/GacA two-component system in *Pseudomonas aeruginosa*. *PLoS Genet.* **12**, e1006032 (2016).
3. Thomas, S.A., Brewster, J.A. & Bourret, R.B. Two variable active site residues modulate response regulator phosphoryl group stability. *Mol. Microbiol.* **69**, 453-465 (2008).
4. Pazy, Y. et al. Matching biochemical reaction kinetics to the timescales of life: structural determinants that influence the autodephosphorylation rate of response regulator proteins. *J. Mol. Biol.* **392**, 1205-1220 (2009).
5. Willett, J.W. & Kirby, J.R. Genetic and biochemical dissection of a HisKA domain identifies residues required exclusively for kinase and phosphatase activities. *PLoS Genet.* **8**, e1003084 (2012).
6. Dutta, R., Yoshida, T. & Inouye, M. The critical role of the conserved Thr247 residue in the functioning of the osmosensor EnvZ, a histidine kinase/phosphatase, in *Escherichia coli*. *J. Biol. Chem.* **275**, 38645-38653 (2000).
7. Kinoshita-Kikuta, E. et al. Functional characterization of the receiver domain for phosphorelay control in hybrid sensor kinases. *PLoS ONE* **10**, e0132598 (2015).
8. Liu, Y. et al. A pH-gated conformational switch regulates the phosphatase activity of bifunctional HisKA-family histidine kinases. *Nat Commun* **8**, 2104 (2017).
9. Solà, M. et al. Towards understanding a molecular switch mechanism: thermodynamic and crystallographic studies of the signal transduction protein CheY. *J. Mol. Biol.* **303**, 213-225 (2000).
10. Brencic, A. et al. The GacS/GacA signal transduction system of *Pseudomonas aeruginosa* acts exclusively through its control over the transcription of the RsmY and RsmZ regulatory small RNAs. *Mol. Microbiol.* **73**, 434-445 (2009).
11. Kay, E. et al. Two GacA-dependent small RNAs modulate the quorum-sensing response in *Pseudomonas aeruginosa*. *J. Bacteriol.* **188**, 6026-6033 (2006).
12. Kong, W. et al. Hybrid sensor kinase PA1611 in *Pseudomonas aeruginosa* regulates transitions between acute and chronic infection through direct interaction with RetS. *Mol. Microbiol.* **88**, 784-797 (2013).
13. Bhagirath, A.Y. et al. Characterization of the direct interaction between hybrid sensor kinases PA1611 and RetS that controls biofilm formation and the type III secretion system in *Pseudomonas aeruginosa*. *ACS Infect. Dis.* **3**, 162-175 (2017).
14. Turner, K.H., Everett, J., Trivedi, U., Rumbaugh, K.P. & Whiteley, M. Requirements for *Pseudomonas aeruginosa* acute burn and chronic surgical wound infection. *PLoS Genet* **10**, e1004518 (2014).
15. Hsu, J.L., Chen, H.C., Peng, H.L. & Chang, H.Y. Characterization of the histidine-containing phosphotransfer protein B-mediated multistep phosphorelay system in *Pseudomonas aeruginosa* PAO1. *J. Biol. Chem.* **283**, 9933-9944 (2008).

REVIEWERS' COMMENTS:

Reviewer #1 (Remarks to the Author):

The authors have resolved the issues. The additional information and texts enhanced the clarity of the paper.

Reviewer #2 (Remarks to the Author):

This manuscript is a revision of the work presented by Francis et al titled "Multiple communication mechanisms between sensor kinases are critical for virulence in *Pseudomonas aeruginosa* ». In this revised manuscript the authors answer to all of the points raised by the reviewers and have carry out all of the complementarities experiment requested. These new experiments strengthen the message of this manuscript. I have only one concern about this revision. In the conclusion the authors state that D858 RetS mutation is essential in PAO1 and PA103 but not in PAK and they try to give an explanation. They propose that this difference is link to a "possible" difference in RetS production "perhaps" linked to a difference in codon usage. I appreciate the effort made by the authors to reconcile the results of different studies. The only way to prove a difference in RetS production or stability could be to perform western blot analysis on RetS protein in the 3 strains to identify a "possible difference" in their production and I don't ask this to the authors. My fear is that some readers will just take the following message "PAK produces less RetS protein than PAO1" and they will use this information as an established fact in their publications to build some hypothesis in their conclusion. That will create mistake in the bibliography. The works presented here are convincing and this manuscript doesn't need a speculation to reconcile the results of different studies. It would be fairer to just state that D858 RetS mutation is essential in PAO1 and PA103 but not in PAK and no more.

Again, this is an excellence work.

We thank the reviewers again for their careful consideration of our manuscript. We have implemented the change that they requested.

Reviewer #1 (Remarks to the Author):

The authors have resolved the issues. The additional information and texts enhanced the clarity of the paper.

Reviewer #2 (Remarks to the Author):

This manuscript is a revision of the work presented by Francis et al titled "Multiple communication mechanisms between sensor kinases are critical for virulence in Pseudomonas aeruginosa ». In this revised manuscript the authors answer to all of the points raised by the reviewers and have carry out all of the complementarities experiment requested. These new experiments strengthen the message of this manuscript. I have only one concern about this revision. In the conclusion the authors state that D858 RetS mutation is essential in PAO1 and PA103 but not in PAK and they try to give an explanation. They propose that this difference is link to a "possible" difference in RetS production "perhaps" linked to a difference in codon usage. I appreciate the effort made by the authors to reconcile the results of different studies. The only way to prove a difference in RetS production or stability could be to perform western blot analysis on RetS protein in the 3 strains to identify a "possible difference" in their production and I don't ask this to the authors. My fear is that some readers will just take the following message "PAK produces less RetS protein than PAO1" and they will use this information as an established fact in their publications to build some hypothesis in their conclusion. That will create mistake in the bibliography. The works presented here are convincing and this manuscript doesn't need a speculation to reconcile the results of different studies. It would be fairer to just state that D858 RetS mutation is essential in PAO1 and PA103 but not in PAK and no more.

Again, this is an excellence work.

We have removed this speculation from the manuscript as advised by the reviewer.